# Novel Brown Coat Color (Cocoa) in French Bulldogs Results from a Nonsense Variant in *HPS3*

**DOI:** 10.3390/genes11060636

**Published:** 2020-06-09

**Authors:** Sarah Kiener, Alexandra Kehl, Robert Loechel, Ines Langbein-Detsch, Elisabeth Müller, Danika Bannasch, Vidhya Jagannathan, Tosso Leeb

**Affiliations:** 1Institute of Genetics, Vetsuisse Faculty, University of Bern, 3001 Bern, Switzerland; sarah.kiener@vetsuisse.unibe.ch (S.K.); dlbannasch@ucdavis.edu (D.B.); vidhya.jagannathan@vetsuisse.unibe.ch (V.J.); 2Dermfocus, University of Bern, 3001 Bern, Switzerland; 3Laboklin, 97688 Bad Kissingen, Germany; kehl@laboklin.com (A.K.); langbein@laboklin.com (I.L.-D.); mueller@laboklin.com (E.M.); 4VetGen, Ann Arbor, MI 48108, USA; loechel@umich.edu; 5Department of Population Health and Reproduction, School of Veterinary Medicine, University of California, Davis, CA 95616, USA

**Keywords:** dog, *Canis lupus familiaris*, whole genome sequence, wgs, heterogeneity, melanosome, pigmentation

## Abstract

Brown or chocolate coat color in many mammalian species is frequently due to variants at the B locus or *TYRP1* gene. In dogs, five different *TYRP1* loss-of-function alleles have been described, which explain the vast majority of dogs with brown coat color. Recently, breeders and genetic testing laboratories identified brown French Bulldogs that did not carry any of the known mutant *TYRP1* alleles. We sequenced the genome of a *TYRP1^+/+^* brown French Bulldog and compared the data to 655 other canine genomes. A search for private variants revealed a nonsense variant in *HPS3*, c.2420G>A or p.(Trp807*). The brown dog was homozygous for the mutant allele at this variant. The *HPS3* gene encodes a protein required for the correct biogenesis of lysosome-related organelles, including melanosomes. Variants in the human *HPS3* gene cause Hermansky–Pudlak syndrome 3, which involves a mild form of oculocutaneous albinism and prolonged bleeding time. A variant in the murine *Hps3* gene causes brown coat color in the *cocoa* mouse mutant. We genotyped a cohort of 373 French Bulldogs and found a strong association of the homozygous mutant *HPS3* genotype with the brown coat color. The genotype–phenotype association and the comprehensive knowledge on *HPS3* function from other species strongly suggests that *HPS3*:c.2420G>A is the causative variant for the observed brown coat color in French Bulldogs. In order to clearly distinguish *HPS3*-related from the *TYRP1*-related brown coat color, and in line with the murine nomenclature, we propose to designate this dog phenotype as “cocoa”, and the mutant allele as *HPS3^co^*.

## 1. Introduction

Melanins are synthesized by melanocytes, and represent pigments in the hair and skin of mammals. Normal pigment formation requires correct melanocyte migration during embryogenesis, correct interaction between melanocytes and other cells, acquisition of the correct cellular and subcellular morphology and the correct activation and function of enzymes [1]. 

Three important enzymes that take part in the melanin biogenesis are tyrosinase (TYR), and the tyrosinase-related proteins 1 and 2 (TYRP1 and TYRP2), which all catalyze redox reactions of pigment precursor molecules. Mammals can produce two different kinds of melanins, the yellow-reddish pheomelanin and the black eumelanin. TYRP1 and TYRP2 are required for normal eumelanin synthesis. A loss of TYRP1 activity leads to the accumulation of brown immature precursors of eumelanin [2,3]. The *TYRP1* gene represents the B locus from classical genetics, and *TYRP1* variants have been described in humans with oculocutaneous albinism type III [4], as well as many animal species with brown coat or feather color, including cats, cattle, chicken, goats, mice, minks, pigs, quail, rabbits and sheep [5,6,7,8,9,10,11,12,13,14].

In dogs, three different variants in *TYRP1* are known to cause brown or chocolate coat color in many breeds [15]. In addition, two younger breed-specific *TYRP1* variants were described in Australian Shepherds [16] and Lancashire Heelers [17]. The corresponding alleles are abbreviated as b^c^ (p.Cys41Ser), b^s^ (p.Gln331*), b^d^ (p.Pro345del), p.Tyr185* and b^e^ (p.Phe342Cys) [15,16,17,18]. In dogs, the wildtype allele *B* leading to black coat color is dominant, whereas the recessive brown phenotype is the result of any combination of two mutant *b* alleles [15]. A splice site variant in the *OCA2* gene was reported in three German Spitz siblings with a light brown coat color in combination with blue eyes and mild photophobia [19].

Dog breeders and diagnostic testing laboratories recently recognized brown French Bulldogs that did not carry any of the known mutant *TYRP1* alleles. We therefore initiated this study with the aim to identify the genetic variant causing this new brown coat color in French Bulldogs.

## 2. Materials and Methods

### 2.1. Ethics Statement

All dogs in this study were privately owned, and samples were collected with the consent of their owners. The collection of blood samples was approved by the “Cantonal Committee For Animal Experiments” (Canton of Bern; permit 75/16).

### 2.2. Animal Selection

This study included 373 French Bulldogs (Appendix A). They included 130 cases with brown or lilac (= dilute brown) base color, 111 controls with black or blue (=dilute black) base color and 132 dogs whose coat color phenotype with respect to brown eumelanin could not reliably be determined. These included fawn, cream and white dogs, as well as dogs for which we could not obtain reliable coat color information from the owners. In most of the fawn, cream and white dogs, it would have been possible to discriminate between black and brown eumelanin, based on the pigmentation of the nose. However, as this requires high quality photographs, which were not available for all dogs, we chose to exclude such dogs from the genotype–phenotype association. Genomic DNA was isolated with standard protocols from EDTA (ethylenediaminetetraacetic acid) blood samples, cheek swabs or hair roots.

### 2.3. Whole Genome Sequencing

An Illumina TruSeq PCR-free DNA library with ~450 bp insert size of a brown French Bulldog was prepared. We collected 149 million 2 × 125 bp paired-end reads or 14× coverage on a HiSeq2500 instrument (Illumina, San Diego, CA, USA). The reads were mapped to the dog reference genome assembly CanFam3.1 and aligned as described [20]. Briefly, after trimming adaptor sequences and low-quality bases at the ends of reads with FASTP [21], BWA version 0.7.13 [22] was used for the alignment to the canine reference genome. Samtools version 0.1.18 [23] was used to sort the aligned reads by coordinates, and to produce bam-files. Duplicates were marked with Picard tools [24]. The sequence data were submitted to the European Nucleotide Archive with the study accession PRJEB16012 and sample accession SAMEA4504835.

### 2.4. Variant Calling

Variant calling was performed using GATK version 3.8 software [25], as described [20]. The main steps of variant calling included base quality recalibration with BaseRecalibrator (within GATK), followed by the actual variant calling with the HaplotypeCaller algorithm of GATK. To predict the functional effects of the called variants, SnpEff [26] software together with NCBI annotation release 105 for the CanFam 3.1 genome reference assembly was used. For variant filtering we used 655 control genomes (Appendix A).

### 2.5. Gene Analysis

We used the dog reference genome assembly CanFam3.1 and NCBI annotation release 105. Numbering within the canine *HPS3* gene corresponds to the NCBI RefSeq accession numbers XM_542830.6 (mRNA) and XP_542830.3 (protein).

### 2.6. Sanger Sequencing

To confirm the candidate variant *HPS3*:c.2420G>A, and to genotype all of the dogs in this study, Sanger Sequencing was used. A 354 bp PCR product was amplified from genomic DNA using AmpliTaqGold360Mastermix (Thermo Fisher Scientific, Waltham, MA, USA) and the primers 5‘-TCTGGGATATGGGGGCTTGA-3′ (Primer F) and 5′-TGCAAGGAATTTACTCATGGACG-3′ (Primer R). After treatment with shrimp alkaline phosphatase and exonuclease I, PCR amplicons were sequenced on an ABI 3730 DNA Analyzer (Thermo Fisher Scientific, Waltham, MA, USA). Sanger sequences were analyzed using the Sequencher 5.1 software (GeneCodes, Ann Arbor, MI, USA).

## 3. Results

### 3.1. Phenotype Characterization

Several brown French Bulldogs were genotyped as homozygous for the wild type allele at all three common *TYRP1* variants (b^c^, b^s^, b^d^). We therefore hypothesized that their coat color was due to a new allele that has not yet been reported in the literature. This new coat color appeared to be slightly darker than the *TYRP1*-related chocolate coat color in adult dogs. From now on, we will refer to this dark brown phenotype as cocoa (Figure 1).

### 3.2. Genetic Analysis

In order to characterize the hypothetical new allele and the underlying causative genetic variant, we sequenced the genome of one cocoa French Bulldog at 14× coverage and searched for homozygous and heterozygous variants that were not present in the genomes of 647 other dogs and 8 wolves (Table 1, Appendix A).

This analysis identified 2 homozygous and 48 heterozygous protein-changing, private variants (Appendix A). The variants were prioritized based on their potential functional impact and the known functions of the respective genes from the literature. We considered a single nucleotide variant in *HPS3* as the most likely candidate causative variant, as variants in *HPS3* lead to brown pigmentation phenotypes in humans with Hermansky–Pudlak syndrome 3 and the *cocoa* mouse mutant [22,23].

The identified canine variant can be designated as Chr23:43,969,695G>A (CanFam3.1) or XM_542830.6:c.2420G>A (Figure 2). This is a nonsense variant predicted to truncate the last 196 amino acids of the wild type HPS3 protein, XP_542830.3:p.(Trp807*). We did not investigate whether any mutant protein is expressed, or whether the premature stop codon leads to nonsense-mediated mRNA decay.

We confirmed the presence of the *HPS3* variant by Sanger sequencing and genotyped a cohort of 372 additional French Bulldogs. The index case and 46 additional dogs of the 130 dogs with an owner-declared brown or lilac (= dilute brown) coat color were homozygous for the mutant allele in *HPS3*. Eighty-two of the remaining 83 cases had two mutant *TYRP1* alleles, which explained their brown coat color. One dog with a dark brown and tan coat color did not carry any of the known mutant alleles at *HPS3* or *TYRP1*.

The *HPS3* mutant allele was not detected in the homozygous state in any of the 111 black or blue French Bulldogs. However, 42 of these dogs carried the mutant *HPS3* allele in a heterozygous state (Table 2, Appendix A). In French Bulldogs with cream, fawn or white coat color, homozygous mutant *HPS3* genotypes occurred in 11 dogs. Since these dogs produce eumelanin only on the nasal planum, the effect of the homozygous mutant *HPS3* genotype is not easily visible.

## 4. Discussion

In this study, we identified a homozygous nonsense variant, *HPS3*:c.2420G>A, as a plausible candidate causative variant for a new brown coat color phenotype in French Bulldogs. Genetic variants in *HPS3* are known to cause Hermansky–Pudlak syndrome type 3 (HPS3) in humans, which is a rare autosomal recessive disorder characterized by oculocutaneous albinism and a bleeding disorder with storage pool deficiency due to the absence of platelet-dense bodies [27,28]. HPS3 patients additionally have mild nystagmus and mildly reduced visual acuity [27]. The phenotype of the homologous *cocoa* mouse mutant, characterized by a brown coat and prolonged bleeding time, is caused by a genetic variant in the murine *Hps3* gene [29,30].

*HPS3* encodes a subunit of a protein complex named Biogenesis of Lysosome-related Organelles Complex-2 (BLOC-2) [31,32]. This protein complex controls the sorting and transport of newly synthesized integral membrane proteins from early endosomes to both lysosomes and lysosome-related organelles (LROs), such as melanosomes and platelet-dense granules. In the case of melanosomes, BLOC-2 interacts with two proteins from the RAB family (RAB32, RAB38), and they likely identify specialized early endosomal domains for the budding of transport intermediates destined for maturing melanosomes [33]. The melanosomes undergo four distinct steps of maturation: Stage I pre-melanosomes are non-pigmented vacuoles that are derived from the endosomal system. These then acquire characteristic internal striations (stage II). Melanin pigment is deposited onto the striations (stage III), eventually giving rise to mature, fully melanized stage IV melanosomes [34]. A malfunctioning BLOC-2 manifests itself in an increase in the percentage of both multivesicular and type II/III forms, and a relative lack of elliptical type IV forms; most fully pigmented melanosomes in mouse strains lacking a component of BLOC-2 are spherical, and most likely represent immature melanosomal forms [35]. It was shown that endosomal trafficking of TYRP1 from endosomes to melanosomes is abnormal in melanocytes deficient in BLOC-2. TYRP1 is then mislocalized and accumulated in early endosomes, instead of being delivered to the melanosomes where melanin synthesis could begin [36,37].

The available knowledge on *HPS3* provides a mechanistic hypothesis for the pigmentation phenotype in cocoa dogs: We speculate that due to the lack of HPS3, melanosome biogenesis is impaired, resulting in melanocytes that have a smaller than normal proportion of fully pigmented mature melanosomes, which might result in a lighter coat color. At the same time, as TYRP1 is not efficiently incorporated into melanosomes, eumelanin synthesis in cocoa dogs may result in the formation of brown eumelanin precursors instead of the mature black eumelanin and also contribute to the phenotype. In contrast to completely TYRP1-deficient (chocolate) dogs, the darker shade of brown in adult cocoa dogs suggests that the synthesis of mature eumelanin is only partially and not completely blocked in cocoa dogs.

Based on the comprehensive knowledge on *HPS3* function in humans and mice, together with the observed genotype–phenotype association in a large cohort of French Bulldogs, we think that *HPS3*:c.2420G>A is very likely the causative genetic variant for the brown coat color in the investigated French Bulldogs. Consequently, we propose to designate the coat color phenotype in these dogs as cocoa to emphasize the locus heterogeneity and to clearly distinguish it from *TYRP1*-related forms of brown coat color. Cocoa in adult dogs appears slightly darker as *TYRP1*-related brown. However, coat colors are also influenced by the genetic background, and it is probably not possible to reliably distinguish these two coat colors without genetic testing. The finding of one brown dog that was a homozygous wildtype at all four tested variants for brown coat color suggests an even more complex heterogeneity and the existence of further, yet uncharacterized causal variants.

Hematological or ophthalmologic examinations were not performed to investigate whether *HPS3* mutant cocoa French Bulldogs have any pathological phenotypes, such as prolonged bleeding time or visual impairment. Additional studies clarifying these open questions are urgently required. Due to the potential animal welfare concern, further breeding of cocoa-colored dogs should only be considered if these dogs do not have any clinically relevant impairments.

## 5. Conclusions

We identified the *HPS3*:c.2420G>A nonsense variant as likely causative for the cocoa coat color in French Bulldogs. The phenotype is inherited as an autosomal recessive trait. Our data enable genetic testing for the cocoa allele in French Bulldogs. Additional studies are warranted to clarify whether cocoa dogs have any bleeding disorders or visual impairment.

## Figures and Tables

**Figure 1 genes-11-00636-f001:**
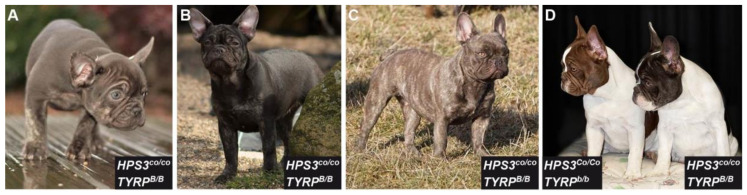
Coat color phenotype of cocoa and chocolate French Bulldogs. Genotypes at the underlying loci are indicated (see Section 3.2) (**A**) Cocoa puppy with brown coat and blue eyes. (**B**) Same dog as shown in (**A**) as an adult. Note that the coat and eye color has markedly darkened over time. (**C**) Cocoa brindled dog. (**D**) *TYRP^b/b^* (chocolate) and *HPS3^co/co^* (cocoa) mutant dogs in comparison. In adult dogs, cocoa is slightly darker than *TYRP1*-related brown. Photo credits: Heike Ulrich, Joyce Wild.

**Figure 2 genes-11-00636-f002:**
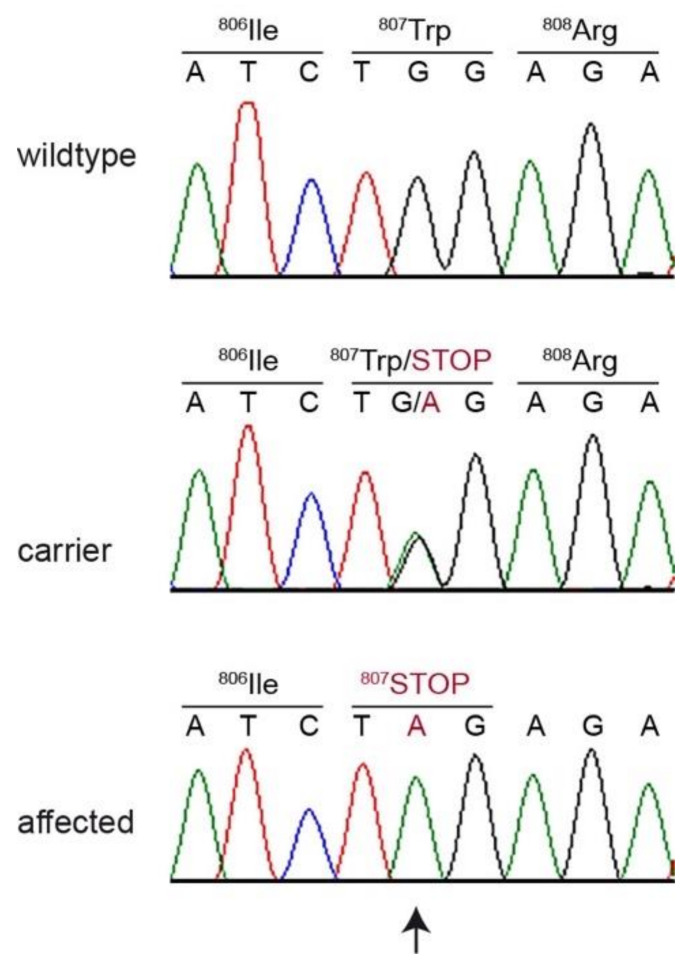
Details of the *HPS3*:c.2420G>A variant. Representative electropherograms of three dogs with different genotypes are shown. The variable position is indicated by an arrow, and the amino acid translations are shown.

**Table 1 genes-11-00636-t001:** Results of variant filtering in a brown (cocoa) French Bulldog and 655 control genomes.

Filtering Step	Homozygous Variants	Heterozygous Variants
all variants	2,571,692	3,132,757
private variants	694	5483
protein-changing private variants	2	48

**Table 2 genes-11-00636-t002:** Genotype–phenotype association of the *HPS3*:c.2420G>A variant in dogs with at least one wildtype *TYRP1* allele (*TYRP1^B/-^*). Detailed information on all phenotypes and genotypes of all 373 studied dogs are listed in Appendix A.

Dogs	G/G	G/A	A/A
Cases (brown or lilac French Bulldogs; *n* = 48) ^1^	1	0	47
Controls (black or blue French Bulldogs; *n* = 111) ^2^	69	42	0
French Bulldogs with other or unknown coat colors (*n* = 96) ^3^	61	24	11

^1^ Cases include uniformly pigmented dogs and brindled dogs with brown or lilac eumelanistic stripes. A total of 85 brown or lilac dogs with *TYRP1^b/b^* genotypes were excluded from this group. ^2^ Black or blue in our study includes dogs with uniform coat colors as well as brindled dogs, where the color of the eumelanistic stripes is black or blue. ^3^ A total of 36 dogs with *TYRP1^b/b^* genotypes were excluded from this group.

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
