# Peer review of "Novel Brown Coat Color (Cocoa) in French Bulldogs Results from a Nonsense Variant in HPS3"

_genes, 2020, doi:10.3390/genes11060636_

Round 1

Reviewer 1 Report

With respect to a brown coat color variation in bulldogs, the authors showed that the responsible gene is not TYRP1 and searched for a novel responsible mutation. The results show that a single nucleotide substitution that produces a stop codon in the HPS3 gene is the most likely causative mutation. There are no comments directed for the revision of the manuscript at all, only that the extra periods in lines 105 and 139 need to be removed. Again, I would like to say that this study is excellent.

Author Response

Extra periods in lines 105 and 139 need to be removed.

Response: Revised accordingly.

Reviewer 2 Report

The aim of submitted article was to identify the genetic variant causing this new brown coat color in French Bulldogs. The paper contribute new, high scientific quality, relevance to the field of dogs` genetic. Every kind of the reliable research results in this field are very needed for sciences, as well as for breeders of dogs.

The Introduction section make enough references to the latest literature on the subject. The research hypothesis and the aim of the research was clearly formulated. The study bring new insights into the relevant field of knowledge.

The Results and Discussion section provide adequate presentation of the authors’ own results as well as the discussion provide proper interpretation of the research in the context of the literature and the conclusions justified by the results obtained. Appropriate methods were used for molecular analysis and they were properly interpreted. Figure 1 significantly enrich the article.

Author Response

No changes requested.

Reviewer 3 Report

The paper provides clear evidence for an interesting novel genetic variant for brown coat color variation of French bulldogs in the absence of TYRP1 and TYRP2 mutations. The paper is a welcome addition to understanding the complex phenotypes represented across Mammalia. I have no major criticism of the paper, and recommend the paper be published after a few minor changes.

The methods section would be improved by including a brief synopsis of methods rather than referring to another paper in the "whole genome sequencing" even if the methods are highly similar. I highly recommend the authors include more details within the context of this paper.

The discussion section would benefit from reworking the second paragraph, as the mechanism described in support of the new phenotype is unclear. Improving the presentation of the pathway relating BLOC-2 and TYRP1 would increase the readability of the manuscript.

Author Response

(1)

The methods section would be improved by including a brief synopsis of methods rather than referring to another paper in the "whole genome sequencing" even if the methods are highly similar. I highly recommend the authors include more details within the context of this paper.

Response: We added more details including the used software and five additional references to the chapters on whole genome sequencing and variant calling.

(2)

The discussion section would benefit from reworking the second paragraph, as the mechanism described in support of the new phenotype is unclear. Improving the presentation of the pathway relating BLOC-2 and TYRP1 would increase the readability of the manuscript.

Response: We expanded the paragraph and formulated a mechanistic hypothesis how deficiency in HPS3 leads to a slightly darker phenotype than deficiency in TYRP1.